# *Sambucus nigra*-Lyophilized Fruit Extract Attenuated Acute Redox–Homeostatic Imbalance via Mutagenic and Oxidative Stress Modulation in Mice Model on Gentamicin-Induced Nephrotoxicity

**DOI:** 10.3390/ph18010085

**Published:** 2025-01-13

**Authors:** Kamelia Petkova-Parlapanska, Ivaylo Stefanov, Julian Ananiev, Tsvetelin Georgiev, Petya Hadzhibozheva, Veselina Petrova-Tacheva, Nikolay Kaloyanov, Ekaterina Georgieva, Galina Nikolova, Yanka Karamalakova

**Affiliations:** 1Department of Chemistry and Biochemistry, Medical Faculty, Trakia University, 11 Armeiska Str., 6000 Stara Zagora, Bulgaria; kamelia.parlapanska@trakia-uni.bg; 2Department of General and Clinical Pathology, Forensic Medicine and Deontology, Faculty of Medicine, Trakia University, 11 Armeiska Str., 6000 Stara Zagora, Bulgaria; ivaylo.stefanov@trakia-uni.bg (I.S.); julian.r.ananiev@trakia-uni.bg (J.A.); ekaterina.georgieva@trakia-uni.bg (E.G.); 3Department “Physiology, Pathophysiology and Pharmacology” Medical Faculty, Trakia University, 11 Armeiska Str., 6000 Stara Zagora, Bulgaria; tsvetelin.georgiev@trakia-uni.bg (T.G.); petya.hadzhibozheva@trakia-uni.bg (P.H.); 4Department of Molecular Biology, Immunology and Medical Genetics, Medical Faculty, Trakia University, 6000 Stara Zagora, Bulgaria; veselina.petrova@trakia-uni.bg; 5Department of Organic Chemistry, University of Chemical Technology and Metallurgy, 8 St. Kliment Ohridski Blvd., 1756 Sofia, Bulgaria; nikolaykaloyanov@uctm.edu

**Keywords:** *S. nigra* extract, acute model, gentamycin, nephroprotection

## Abstract

**Background:** Gentamicin (GM) administration is associated with decreased metabolism, increased oxidative stress, and induction of nephrotoxicity. *Sambucus nigra* L., containing flavonoids, anthocyanins, and phytosterols, possesses antioxidant and anti-inflammatory potential. **Objectives:** The present study aimed to investigate the nephroprotective and anti-inflammatory potential of lyophilized *Sambucus nigra* fruit extract (*S. nigra extract*) to reduce acute oxidative stress and residual toxicity of GM in a 7-day experimental model in Balb/c rodents. **Methods:** The *S. nigra* extract was lyophilized (300 rpm; 10 min; −45 °C) to improve pharmacological properties. Balb/c mice were divided into four (n = 6) groups: controls; *S. nigra* extract per os (120 mg kg^−1^ day^−1^ bw); GM (200 mg kg^−1^ day^−1^ bw) (4); and GM + *S. nigra* therapy. The activities of antioxidant and renal enzymes, cytokines, and levels of oxidative stress biomarkers—Hydroxiproline, CysC, GST, KIM-1, PGC-1α, MDA, GSPx—were analyzed by ELISA tests. The ROS and RNS levels, as well as 5-MSL-protein oxidation, were measured by EPR spectroscopy. **Results:** The antioxidant-protective effect of *S. nigra* extract (120 mg kg^−1^) was demonstrated by reduced MDA, ROS, and RNS and increased activation of endogenous enzymes. Furthermore, *S. nigra* extract significantly reduced the expression of IL-1β, IL-6, IL-10, TNF-α, IFN-γ, and KIM-1 and regulated collagen/protein (PGC-1α and albumin) deposition in renal tissues. **Conclusions:** Histological evaluation confirmed that *S. nigra* (120 mg kg^−1^) attenuated renal dysfunction and structural damage by modulating oxidative stress and acute inflammation and could be used as an anti-fibrotic alternative in GM nephrotoxicity.

## 1. Introduction

Gentamicin (GM), as a broad-spectrum aminoglycoside antibiotic, has marked antibiotic activity against aerobic/Gram-negative bacterial infections. Clinically, GM is used in the systemic treatment of septicemia, nosocomial infections of the respiratory tract, urinary tract, and intra-abdominal infections [1]. Despite its application, GM is toxic due to reduced metabolism and difficulty in excretion with normal renal function [1,2]. The cellular toxicity pathways by which GM treatment leads to cellular destruction is concentration in the Golgi complex, followed by concentrations in the endosomal and lysosomal vacuoles of the proximal tubular cells of the kidney, leading to inflammatory and vascular responses, and acute tubular necrosis [1,2]. Different hypotheses indicate that such toxicity appears to be induced by abnormal vesicle fusion, reduced protein synthesis, and increased mitochondrial toxicity, followed by an increased imbalance of aerobic oxidative stress (OS) [3]. Directly or indirectly, the cytoplasmic GM accumulation leads to OS induction, followed by mitochondrial activity reduction and apoptotic activation [3]. GM therapy causes OS by increasing the production of reactive oxygen and nitrogen species (ROS/RNS), inflammation, and fibrosis [3,4]. The short-lived ROS (superoxide ion (•O_2_**^−^**), hydrogen peroxide (H_2_O_2_)), RNS (nitric oxide (•NO), peroxynitrite (ONOO−)), and the oxidized lipids as redox signaling agents generated under the cytokines and enzymes control (nitric oxide synthase, lipoxygenases and cyclooxygenases) [4] are directly involved in the metabolic regulation and adaptation to xenobiotic stress [5]. Afterward, in healthy cells, the redox balance is rapidly reduced by enzymatic and non-enzymatic systems (glutathione (GSH), melatonin, uric acid, bilirubin, and vitamin C) by natural ferro-statins, such as vitamin E, and by plant polyphenols [6,7,8]. The inability of the intracellular antioxidants to deal with oxidative disturbances leads to free radical’s induction and oxidative cascade signaling [7,8] and is involved in renal, cardiovascular, hematological, and inflammatory pathogenesis [8,9,10,11,12]. Moreover, GM therapy reduces enzyme efficacy, impairs lipid peroxidation [10], and induces renal genotoxicity by incensement of ROS/RNS [10].

Overall, plant compounds with antioxidant, anti-inflammatory, and radical-scavenging properties play a significant protective role in direct renal protection and the ability to ameliorate GM-induced nephrotoxicity [13,14,15]. Recently, there has been much interest in investigating the effects of plants containing polyphenols and quercetin-like flavonoids as protectors of inflammatory activation. The polyphenols and flavonoids reduce ROS and RNS production as controlled oxidative stress changes in acute kidney injury [16,17,18,19].

*Sambucus nigra (S. nigra*, black elderberry) is an active plant with antioxidant, antibacterial, and antitumor properties [20,21,22]. Traditionally, in the Balkans, the most popular is the usage of *S. nigra* fruits, with high content of flavonoids (rutin, quercetin-3-*O*-rutinoside, kaempferol-3-*O*-rutinoside), anthocyanins, phytosterols, triterpenes, tannins, glycosides, *p*-coumaric acid, and lectins, which determine the anti-inflammatory and immunomodulatory promoting activities [23,24,25,26]. The antioxidant properties of *S. nigra* extract have been verified to primarily depend on the presence of rutin, quercetin-3-*O*-rutinoside, anthocyanins, and other phenolic compounds. Furthermore, the presence of flavonoids and phenolic compounds in *S. nigra* helps inhibit acute phase inflammation, modulating renal toxicity through increased enzymatic protection and alleviating ROS and RNS accumulation [23,24,25,26]. In addition, it has been proven that ROS regulation by *S. nigra* is linked to the prevention of GM-induced renal intoxication and necrosis [25,26,27,28]. Previous research has shown that rutin and epigallocatechin contending *S. nigra* extracts reduced lipid peroxidation [10,11,26]. Interestingly, it has been demonstrated that rutin contending *S. nigra* extracts ameliorate nitric oxide (NO_2_) and nitric radicals (•NO) and diminish pro-inflammatory cytokine signaling as tumor necrosis factor (TNF) and interleukins IL-1β and IL-6 [29]. The experimental research on cell cultures and animal models shows that *S. nigra* fruit extracts promoted the increased ferocytosis of apoptotic neutrophils by increasing IL-10, reduced TNF and heme oxygenase-1, i.e., modulated redox balance by stimulating the antioxidant and enzymatic production [26,29]. Recently, it has been shown that *Sambucus sp.* reduced OS and inflammatory stimulated preadipocytes and macrophages by cyto-protective [26] and geno-protective properties.

Numerous components of *S. nigra* fruits have been identified, but their biological activity is still not fully understood. The use of dietary supplements and preparations of *S. nigra* fruits is accompanied by a lack of complete information on the composition, dosage, anti-inflammatory and protective activity, and compatibility with other preparations. Extracts of *S. nigra* fruits have not been extensively studied for their radical-modulating and anti-inflammatory effects in vivo in animal models.

Therefore, we investigated the cytoprotective and antioxidant therapeutic effects of a lyophilized *S. nigra* fruit extract on GM-induced nephrotoxicity in a 7-day experimental rodent model. Furthermore, we focused on the possible oxidative mechanisms of action of *S. nigra*-lyophilized extract against acute renal toxicity. We suggest that the ameliorating effect of the lyophilized extract of *S. nigra* is focused on protective action and direct regulation of ROS, NO• and •O_2_^−^ concentrations, redox–homeostatic albumin imbalance, and anti-inflammatory activity in acute GM-nephrotoxicity.

## 2. Results

### 2.1. Phenolics Quantitative Evaluation and Antioxidant Capacity in S. nigra-Lyophilized Extract

The HPLC–DAD analysis showed that rutin (782.6 ± 0.68 µg g^−1^) was the dominant phenolic component in *S. nigra*-lyophilized extract, followed by epigallocatechin (314.15 ± 0.59 µg g^−1^), myricetin (136.06 ± 0.84 µg g^−1^), quercetin (37.9 ± 1.27 µg g^−1^), and other compounds in low concentrations (Table 1).

In addition to Table 1, Figure 1a,b presents the HPLC–DAD chromatograms of the analyzed *S. nigra* extract, compared to standards.

### 2.2. S. nigra-Lyophilized Extract Increased Antioxidant, Anticlastogenic, and Cytoprotective Effects

In order to assess the DPPH/R scavenging, the elevated 0.0691 µg mL^−1^ (87.59 ± 0.037%) of *S. nigra* extract confirms increased antioxidant activity, compared to other published results: 0.049 µg mL^−1^ (62.56 ± 1.12%) scavenging activity of *S. nigra* fruits methanolic extract, and 0.051 µg mL^−1^ (65.965 ± 0.003%) for analyzed fruit samples. Our lyophilized *S. nigra* extract showed a ~23% increase in scavenging potential (Figure 2).

The chromosomal aberrations frequency shows the cytoprotective (*anticlastogenic*) effect of *S. nigra*-lyophilized extract in lymphocyte cultures (Figure 3). In the lymphocytes treated with *S. nigra* extract (n = 300), the aberrant cell percentage was two-fold reduced (0.7%) compared to untreated cells (2.0%) (Cramer’s V = 0.058; *p* < 0.2). A non-aberrant lymphocyte cultures frequency treated with *S. nigra* was 99.3% vs. untreated 98.0% lymphocytes. Despite the weak relationship between the factor “type of lymphocyte treatment” and aberrant cells between the two studied groups, no statistically significant difference was found (*p* = 0.155; *p* > 0.05). It should be noted that the 4 µL mL^−1^
*S. nigra* administration does not display clastogenic activity and protect against oxidative stress (Figure 3).

### 2.3. S. nigra Extract Attenuated GM-Induced Kidney Hypertrophy

Figure 4 shows the *S. nigra* effect on general physical condition, weight gain, and relative kidney weight in BALB/c mice after a 10-day GM induction. Daily GM intoxication resulted in a significant reduction in body weight (~24.63%, *p* < 0.05, *t*-test) and characteristic renal hypertrophy, as assessed by a two-fold increase in kidney weight (0.49 ± 0.02 g vs. 0.25 ± 0.01 g vs. controls). Interestingly, GM-treated mice receiving a daily protective dose of 120 mg kg^−1^ *S. nigra* showed a significant increase in body weight. *S. nigra* extract stimulation improved renal hypertrophy and restored kidney function compared to GM administration, observable by the kidney weight decrease (0.31 ± 0.03 g, *p* < 0.05).

### 2.4. S. nigra Extract Ameliorated Renal Histopathological Changes

Significantly pronounced degenerative, hypermeritic, inflammatory, and vascular changes in renal tissue were registered in the GM group. Pathomorphological renal changes were not observed in the controls, *S. nigra* alone, GM + *S. nigra* combined therapy (Table 2).

### 2.5. S. nigra Extract Ameliorated Renal Fibrosis and Reducde MCs Density and CFT

The beneficial *S. nigra* role in GM-induced renal fibrosis reduction by influencing MCs density was present. The kidney sections from controls, GM, and GM + *S. nigra*-treated mice by both toluidine-blue and Azan staining techniques were examined to determine whether GM-induced kidney fibrosis was reduced after *S. nigra* treatment (Table 3). Significant differences in CFT vs. controls and GM + *S. nigra* mice were not detected.

Metachromatic MC localization was found in the renal corpuscles and between the proximal and distal convoluted tubules on nephron in the renal cortex. Single MCs were estimated in glomeruli and between nephron tubules in controls (Figure 5’).

In GM + *S. nigra* group, MCs densities in mentioned renal structures were not significantly higher, close to controls, and significantly lower in the GM group. In the upper medulla, the MC number in the GM group was four-fold higher compared to the controls and GM + *S. nigra* group. It is important to note that there was not a statistical significance in MCs density between controls and GM + *S. nigra* therapy. In the upper medulla of all treated groups, MCs were located between nephron tubules and collecting ducts. MCs were present in the GM group but absent in controls and GM + *S. nigra* therapy. Therefore, *S. nigra* administration caused a significant MC decrease in the renal cortex and medulla vs. GM group. Azan staining allowed for the detection of CFT and bundles in the renal cortex and medulla in order to define collagen deposition (Figure 5”).

The micro-morphometric study was used to estimate the thickness of collagen fibers and bundles in the renal cortex and medulla. In the renal cortex and upper medulla of the GM group, collagen fibers and bundles between nephron tubules were significantly thicker vs. controls and GM + *S. nigra* combined therapy.

### 2.6. S. nigra Extract Ameliorated Renal Hydroxyproline Content and Protein Oxidation

Acute GM-nephrotoxicity significantly increased HYP content compared to controls (875.18 ± 111.4 mg g^−1^ vs. 425.71 ± 72.9 mg g^−1^, *t*-test, *p* < 0.05). Figure 6A presents a protective *S. nigra* effect against GM renal injury. The hydroxyproline content was statistically significantly decreased in the GM + *S. nigra* group (875.18 ± 111.4 mg g^−1^ vs. 595.09 ± 88.1 mg g^−1^ tissue, *p* < 0.005). The 10 days of protection with *S. nigra* extract statistically insignificant reduced collagen deposition versus controls (401.22 ± 35.4 mg g^−1^ vs. 425.71 ± 72.9 mg g^−1^).

In vivo, remodeled protein oxidation was determined after 5-MSL-albumin/protein conjugation (Figure 6B). Compared with controls, renal protein expression was significantly increased (0.361 ± 0.04 vs. 1.002 ± 0.52 a.u., *p* < 0.005, *t*-test) after GM intoxication. *S. nigra* treatment significantly reduced GM-induced kidney protein dysregulation compared to the GM group (0.409 ± 0.07 vs. 1.002 ± 0.52 a.u., *p* < 0.002, *t*-test). Notably, *S. nigra* administration resulted in inhibition of renal protein expression and reduced OS, with a value comparable to controls (0.41 ± 0.03 vs. 0.361 ± 0.04 a.u., *p* < 0.05, *t*-test).

### 2.7. S. nigra Extract Ameliorated Renal and Serum Injures

Data analysis of renal and blood concentrations of KIM-1, cystatin C, GST, gamma-GT, Cre, and urea in control, *S. nigra*, GM + *S. nigra* groups were performed to detect the *S. nigra* extract ability to protect against acute kidney injury (Figure 7). GM-induced tubular injury and nephrotoxicity resulted in a significant increase in renal KIM-1 expression vs. controls (6.73 ± 0.21 ng/mL vs. 2.56 ± 0.1 ng/mL, *t*-test, *p* < 0.05), the *S. nigra* stimulation (6.73 ± 0.17 ng/mL vs. 2.61 ± 0.09 ng/mL, *t*-test, *p* < 0.005), and the GM + *S. nigra* combination (6.73 ± 0.21 ng/mL vs. 3.86 ± 0.37 ng/mL, *t*-test, *p* < 0.05) (Figure 7A). No significant differences were reported between the controls and after *S. nigra* extract protection.

GM treatment vs. controls had statistically significantly increased renal Cys C expression (0.866 ± 0.07 ng/mL vs. 0.301 ± 0.016 ng/mL, *t*-test, *p* < 0.05; Figure 7B) and increased GST (892.12 ± 55.14 nmol/gPr vs. 512.3 ± 58.59 nmol/gPr, *t*-test, *p* < 0.001; Figure 7C). Also, an incensement was measured in serum Cre (1.86 ± 0.23 ng/dL vs. 0.73 ± 0.1 ng/dL, *p* < 0.005), urea (104.38 ± 22.35 mg/dL vs. 64.82 ± 15.35 mg/dL, *p* < 0.005) (Figure 7D,E), and gamma-GT (2.11 ± 0.09 ng/mL vs. 1.43 ± 0.012 ng/mL, *p* < 0.005, Figure 7F), confirming acute renal injuries, oxidative changes, and neutrophil cells death.

Renal enzyme activity in the GM-administrated group was statistically significantly higher versus GM + *S. nigra* combined therapy. The induction of Cys C (0.866 ± 0.07 ng/mL vs. 0.67 ± 0.04 ng/mL, *t*-test, *p* < 0.05, Figure 7B), GST (892.12 ± 55.14 nmol/gPr vs. 647 ± 33.59 nmol/gPr, *t*-test, *p* < 0.005, Figure 7C), sera creatinine (1.86 ± 0.23 vs. 0.84 ± 0.3, *p* < 0.005, Figure 7D), sera urea (104.38 ± 22.35 vs. 76.85 ± 11.44, *p* < 0.05; Figure 7E), and sera gamma-GT (2.11 ± 0.09 vs. 1.75 ± 0.04, *p* < 0.05, Figure 7F) were significantly reduced in renal tissue and sera after *S. nigra*-lyophilized extract protection. Significant enzymatic activity was not reported between the controls and *S. nigra*-stimulated animals.

### 2.8. S. nigra Extract Normalize Renal Enzyme Activities and Ameliorated Lipid Peroxidation

Next, we explored the effects of GM administration on the renal enzymatic activities. The renal SOD (1.291 ± 0.35 vs. 4.74 ± 0.86 IU/gHb, *p* < 0.005; Figure 8A), CAT (1.62 ± 0.13 vs. 3.78 ± 0.05 IU/gPr, *p* < 0.003; Figure 8B), GPx1 (18.9 9 ± 2.86 vs. 68.84 ± 5.39 IU/gPr, *p* < 0.05, Figure 8C), and GSH (21.99 ± 3.22 vs. 59.85 ± 2.376 IU/gPr, *p* < 0.04, Figure 8D) significantly decreased versus controls, which indicates an impairment in kidney functions. However, GM considerably increased renal MDA concentration (5.87 ± 0.9 vs. 3.09 ± 0.4 µmol/mL, *p* < 0.05; Figure 8E) versus controls.

Conversely, the *S. nigra* co-administration at 120 mg kg^−1^ showed significant GM inhibition and substantial incensement in the enzymatic defense by OS amelioration in kidneys. Markedly, induced activities of SOD (1.291 ± 0.35 vs. 3.81 ± 0.41 IU/gHb, *p* < 0.05), CAT (1.62 ± 0.13 vs. 3.17 ± 0, 19 IU/gPr, *p* < 0.05), GPx1 (18.99 ± 2.86 vs. 50.75 ± 3.39 IU/gPr, *p* < 0.05), GSH (21.99 ± 3.22 vs. 46.97 ± 5.39, *p* < 0.001), and MDA (5.87 ± 0.9 vs. 4.041 ± 0.12 µmol/mL, *p* < 0.05) were registered vs. controls (Figure 8A–D).

### 2.9. S. nigra Extract Ameliorated ROS, NO• u •O_2_^−^ Stress Levels

To confirm the protective role of *S. nigra* extracts against GM-induced renal oxidative stress, the redox-modulated activity was investigated. Significantly, GM treatment increased ROS (2.77 ± 0.35 vs. 0.89 ± 0.06 a.u., *p* < 0.005; Figure 9A), NO• (55.61 ± 5.35 vs. 14.88 ± 1.41 a.u., *p* < 0.002; Figure 9B), and •O_2_^−^ levels (3.4 1 ± 0.72 vs. 0.97 ± 0.09 a.u., *p* < 0.005, Figure 9C) in the renal cortex compared to controls. The highest protection of *S. nigra* extract was observed in the renal cortical ROS reduction (2.77 ± 0.35 vs. 1.36 ± 0.11 a.u., *p* < 0.05) and in NO• (55.61 ± 5.35 vs. 33.7 ± 2.09 a.u., *p* < 0.002) and •O_2_^−^ (3.41 ± 0.72 vs. 1.994 ± 0.12 a.u., *p* < 0.005) concentrations. These findings highlight the *S. nigra* protective effects on renal OS compared to GM-treated mice. Interestingly, *S. nigra* extract co-administration alleviated renal cortical oxidative stress and promoted exogenous–endogenous enzymatic defense by restoring the redox–homeostatic imbalance versus the GM-intoxicated group.

### 2.10. S. nigra Extract Protected the Kidney Against GM-Induced Acute Inflammation

Furthermore, we assessed the influence of *S. nigra* on GM-induced inflammation. The oxidative stress promotes renal damage and provokes fibrotic processes and loss of renal function by concentration-inactivating collagen deposition (PGC-1α) and inflammatory expression (Figure 10A–F). Significantly, GM administration reduced PGC-1α (65%, *p* < 0.001; Figure 10A) deposition and induced the release of inflammatory IL-1β (78.3%), IL-6 (27.4%), IL-10 (29.5%), TNF-α (55.3%), and IFN-γ (41.9%) versus controls (*p* < 0.005).

*S. nigra* extract co-administration potentially rescues renal cells and modulates cellular response by increasing PGC-1α deposition and suppressing inflammation induced by GM intoxication (*p* < 0.01). Furthermore, *S. nigra* extract stimulates renal adaptive immunity and regulates the inflammatory response, almost comparable to controls (*p* < 0.05).

### 2.11. Correlation Dependences Between Parameters for Protective Properties of S. nigra After GM Intoxication

The results of the established correlations between collagen deposition, lipid peroxidation, cytokine expression and markers of oxidative stress in relation to the *S. nigra* extract action of against GM nephrotoxicity are presented in Table 4.

## 3. Discussion

Receptor-mediated endocytosis via the multi-ligand receptors megalin and tubulin promotes GM deposition in renal proximal tubules, altering lysosomal aggregation, phospholipid metabolism, and mitochondrial toxicity [3,13], thus promoting ROS, RNS accumulation, and oxidative stress [30]. Aberrant mitochondrial modulation of ROS and RNS by metabolic, hormonal, and pro-inflammatory factors, as well as endogenous–exogenous antioxidant deactivation [6,7], caused OS and organ changes [31]. GM stimulates the mitochondrial respiratory chain to accumulate H_2_O_2_ synthesis, i.e., •O_2_−, ^1^O_2_, HO•, OH^–^, NO•, and peroxy radicals. GM-induced acute renal inflammation and accumulation of abnormalities in the renal tubules and glomeruli is a consequence of the activation of inflammatory cells and physiologically impaired redox homeostasis, which predisposes to increased lipid peroxidation [7], chromosomal aberrations, and protein denaturation [31,32,33].

Biologically active flavonoids and polyphenols support acute nephron protection due to antioxidant and anti-inflammatory effects and decreased H_2_O_2_-mediated Fe^2+^/Fe^3+^ mobilization from mitochondria [7,24,32,33]. Based on these facts, we aimed to establish the protective effect of *S. nigra* extract against acute kidney injury and renal lesions directly caused by GM therapy. In addition, we investigated the *S. nigra* antioxidant regulative mechanism by maintaining the redox–homeostatic imbalance and cytoprotective effect, leading to protein oxidation remodeling and cytokine inflammation.

Ultrasonic extraction at 80 kHz improved the extraction yields of rutin (8754 μg mL^−1^) and epigallocatechin (3514 µg mL^−1^) (in the maximum amount) [12], the main flavonoids in used *S. nigra* extract. Rutin, epigallocatechin, and myrcetin flavonoids scavenged ROS and RNS and powerfully modulated oxidative stress changes under GM accumulation [24,31]. Previous studies have demonstrated significant amounts of polyphenolic metabolites deposited in renal tissue after a polyphenolic diet after GM-nephrotoxicity [34,35,36,37]. Rutin and epigallocatechin-rich extracts obtained from *Opuntia ficus Indica*, *Morus alba* L., and *S. nigra* had a nephroprotective effect [38,39,40,41]. The antioxidant capacity, recording 86.5 ± 0.037% ROS-inactivation in vitro, is probably due to the high rutin > epigallocatechin > myricetin > quercetin amounts. The relationship between phenolic content and antioxidant activity of *S. nigra* extract [24] was confirmed. Putri et al. [42] provided evidence that *S. nigra* extracts were characterized by antioxidant activity, reaching 89.25% concerning DPPH radicals responsible for the H_2_O_2_ detoxification and renal function improvement [43].

The cytoprotective efficacy test, which directly assesses the inhibition of genotoxicity, mutagenesis, and carcinogenesis by flavonoid-containing plants [44], attracted our attention. *S. nigra* extract (4 µL mL^−1^) showed a non-toxic, protective effect against lymphocyte cultures without exhibiting clastogenic activity compared to controls. Furthermore, *S. nigra* treatment reduced by 0.7 ± 0.02% (Cramer’s V = 0.058; *p* < 0.2) the deviation rate, demonstrating adequate antigenicity in lymphocytes. Genotoxic mutations induced at cytogenetic levels represent the first step toward carcinogenesis. Understanding the chromosomal mechanisms of action of *S. nigra* suggests a stable antimutagenic potential associated with an enhanced cytoprotective activity [44], minimizing the GM side effects. Furthermore, elderberry flavonoids modulate specific and non-specific immune responses and cytoprotective effects in various acute renal injury models [45].

Mice pretreatment with 120 mg kg^−1^ *S. nigra* extract reduced the initial degenerative, hypermeric, inflammatory, and vascular changes in renal tissue after GM intoxication [12,46]. This is consistent with reduced serum levels at Cre, gamma-GT, urea, and Cys C and significantly improved renal function, eliciting a reduction in the chain inflammatory response. *S. nigra* extract reduced the extracellular matrix and cell proliferation by reducing fibrotic areas in the renal cortex and medulla. Herbal antioxidants reduced GM accumulation by direct GM cytotoxicity mitigation, collagen reduction, suppression of vasoconstriction, and antioxidant–anti-inflammatory action [45]. This motivated us to conduct MCs density, taking into account their role in GM-accumulated fibrosis and to clarify the modulated *S. nigra* effect. *S. nigra* administration alone or in a GM + *S. nigra* combination statistically significantly ameliorates MCs and collagen deposition in the renal cortex (fourfold) and medulla and terminates renal fibrotic, probably by redox-modulative mechanism. Several studies reported that *S. nigra* exerts enhanced antioxidant activity even in patients with idiopathic nephrolithiasis without altering urine pH or H^+^ concentrations [47].

GM increases connective tissue volume and renal fibrosis by increasing TGF-β production, myofibroblast activation, and epithelial–myofibroblast transdifferentiation. Fibroblast TGF-β receptors activate both the Smad pathway and fibroblast collagen expression [48,49] and induce an increased amount of detectable MCs-TGF-β in the initial phase [50] under GM nephrotoxicity. MCs influence fibroblast activity by releasing TGF-β1 and TNF-α mediators [51,52]. The present study reports that *S. nigra* extract directly suppresses the initial phase of GM nephrotoxicity and renal inflammatory response, reducing TGF-β1 and TNF-α by inactivating MCs and collagen proliferation. Similarly, the C-6 ring structure of flavonoids has been reported to inhibit the expression of TGFβ/Smad and TNF-α and regulate collagen synthesis. The flavonoid ring structure suppresses the free radical overexpression [52], and matrix deposition is terminated. Together, rutin, epicatechin, and quartzetin have been shown to reduce ^•^O_2_^−^, H_2_O_2_, OH^•^, and ONOO^−^ cascade in the mitochondrial respiratory chain by enzyme reactivation and fibrotic deactivation. However, flavonoid-containing extracts of *S. nigra* restored TGF-β1, TNF-α, IL-6, serum urea, and Cre [53] to controls through appropriate antioxidant modulation after renal GM disorders [54].

This study focused on identifying the possible mechanisms of *S. nigra* actions on the transmembrane tubular protein KIM-1 responsible for proximal tubular injury [55] and 5-MSL conjugating albumin accounts for conformational changes in –SH in the albumin molecule and remodeling of protein oxidation [56]. GM treatment significantly increased renal KIM-1 expression and structurally changed -SH albumin groups, consistent with previous studies [27,55]. Interestingly, *S. nigra* treatment attenuated KIM-1 expression and protein oxidation to levels comparable to the controls. Acute kidney injuries are alleviated by herbal nutrients through suppressed KIM-1 [57,58] in animals. *S. nigra* extract exhibits a protective effect in kidney injuries [57,58] due to its high protein levels and seven essential amino acids [32,33,43,54], which probably directly inhibits the •O_2_^−^, H_2_O_2_, and HO• production and regulates albumin oxidation [43,59]. In parallel, the urinary albumin–creatinine ratio is reduced by attenuating proximal tubular injury and renal fibrosis [60,61]. Probably, *S. nigra* promotes the quinones production and ability to form irreversible complexes with proteins (*adhesins*), i.e., alters the cellular membrane potential (*hyperpolarization*) and modulates immune decrease in KIM-1, Cys C, and TNF-α in kidneys [61,62]. Seven-day oral exposure to *S. nigra* extract resulted in changes in redox homeostasis, enhanced antioxidant response to increased lipid peroxidation, and inhibition of oxidized albumin -SH groups (*cysteine residues*), suggesting renal mitochondrial and cytoplasmic H_2_O_2_ detoxification [60,62]. Lee et al. [63] proved that elderberry extract containing flavonoids and phytochemicals acts as an intracellular antioxidant, protecting proteins and enzymes from H_2_O_2_-induced ROS and oxidative stress.

Oxidative stress disrupts endogenous and exogenous homeostasis, which is critical in GM-induced renal pathogenesis. Free radicals promote lipid peroxidation in the kidneys, trigger pro-inflammatory pathways, and activate cytokines [10,11,24]. The fact that in the GM-treated group, the MDA and ROS products were almost two-fold increased, but SOD, CAT, and GPx1 enzymes were two times decreased is another sign of GSH depletion and nephrotoxicity activation. Our findings are consistent with previous studies [7,24,63]. In contrast, 120 mg kg^−1^ protection with *S. nigra* extract, both alone and in the GM + *S. nigra* combination, changes hyperpolarization and restores antioxidant homeostasis after a possible respiratory chain reversal to controls. *S. nigra* extract has metal-chelating properties and prevents the initiation of •OH− radicals, resulting in the termination of membrane lipids and DNA fragmentation [7,64]. Therefore, *S. nigra* restores SOD, CAT, and GSH enzymes by further activating GPx1, leading to H_2_O_2_ and lipid peroxide (L-OOH) detoxification. Additionally, extract limits the hydroperoxide synthesis from fatty acids, which allows for the restoration of renal anti-lipid peroxidation and GSH inhibition, even at the peroxisomal level [24,65,66]. *S. nigra* extract containing flavonoids activates a compensatory reaction of the body’s antioxidant defense and improves redox homeostasis by directly mitigating renal toxicity [66,67,68] and fibrotic [69,70].

Exogenous cell-permeable spin probes based on aminoxyl, nitronylnitroxide, or hydroxylamine radicals (CPTIO.K, CMH) permit ROS and RNS determination and directly reflect differences in redox status in vivo [71]. Redox-sensitive agents are reduced to the corresponding diamagnetic forms and act as catalysts in the disposal of renal NO• to the formation of an imino–nitroxide radical or the dismutation of renal •O_2_^−^ to H_2_O_2_ and O_2_ [72,73]. In this context, GM penetrates the membrane by an oxygen-dependent transport mechanism [74] and reduces NO bioavailability in the renal cortex and medulla in harmony with upregulated KIM-1, CysC, and TNF-α activations [75]. Whereas *S. nigra* extract (120 mg kg^−1^) alleviated GM production of NO• and •O_2_^−^ by reducing radicals and normalizing renal cytosolic and mitochondrial SOD and GSH deficits. The extract containing rutin, epigallocatechin, and quercetin prevents the •O_2_^−^ transformation into ONOO¯ and •OH radicals, simultaneously restoring the function of NO• and NO as a signaling molecule regulating blood flow in the renal cortex. Likewise, *S. nigra* stimulates SOD synthesis [24,25,26] and inhibits the xanthine oxidase enzyme [9,67]. Protection (7 days) with the lyophilized extract shows significant catalysis of the •O_2_^−^ and NO• into ONOO¯ radicals by reducing renal stress, followed by an antigen-specific immune response that activates the mitochondrial respiratory chain and reduces GM-induced collagen deposition and cell-mediated fibrotic [13,67]. In accordance, Olas et al. [62] demonstrated that flavonoids, isoflavonoids, and saponins promoted renal blood flow, increased glomerular filtration, and remodulated the Th2 immune response by reducing the expression of inducible NO synthase (iNOS). Therefore, the quinones production by *S. nigra* [73] alters the renal •O_2_^−^ and •NO redox-microenvironment by promoting modifications in redox-sensitive amino acids (*proline*), regulating the selectivity for -SH modifications, i.e., ending renal stress deposition [74,76,77,78] (Figure 1).

GM-induced oxidative stress acutely activates NF-κB pathways, which suppress mitochondrial biogenesis and stimulate the release of pro-inflammatory cytokines in the peripheral renal tissues. PGC-1α, as a coactivator protein, controls mitochondrial biogenesis and adaptive cellular thermogenesis, directly characterizing renal disorders [79].

The presented results show that GM presumably increased the production of IL-1β, IL-6, and IL-10, including TNF-α and IFN-γ. Early-phase inflammatory mediators significantly upregulated PGC-1α. Therefore, 7-day GM accumulation inhibited PGC-1α levels and impaired mitochondrial synthesis, which exacerbated renal stress and promoted nephrotoxicity progression [79,80]. We hypothesize that treatment with lyophilized extract of *S. nigra* regulates Nrf2 signaling pathways by provoking an antioxidant response (SOD, CAT, GPx1, including localized peroxidase and oxidoreductases) against acute-phase inflammatory mediators. In a compositional ratio containing polyphenols, *S. nigra* extract has the potential to suppress ROS, NO•, and •O_2_^−^ generations directly related to lipid peroxidation, i.e., sharply reduces electron transport [81], in the mitochondrial membrane and increases the PGC-1α activity [82]. Presumably, the 7-day-lasting antioxidant compensatory response of *S. nigra*, containing rutin, epigallocatechin, and quercetin, stimulates Th1/Th2 response and protects renal mitochondrial dysfunction by improving renal tubular dynamics and arresting the fibrosis process [83]. Moreover, the potential of the synergistic phytochemicals rutin, epigallocatechin, quercetin, and myricetin in *S. nigra* extract is expressed in the complete antagonism/HO•, •O_2_^−^, NO• scavenging, inhibition of lipid peroxidation, and complete oxidative stress modulation. It has been shown that the synergistic phytochemicals functionally have a ROS/PNH scavenging capacity 100–300 times higher than the mannitol [10,84]. *S. nigra* extract disrupts extracellular matrix components that allow for full penetration of antibiotics, enhance their effect, and simultaneously prevent the accumulation of acute renal dysfunction.

**Scheme 1 pharmaceuticals-18-00085-sch001:**
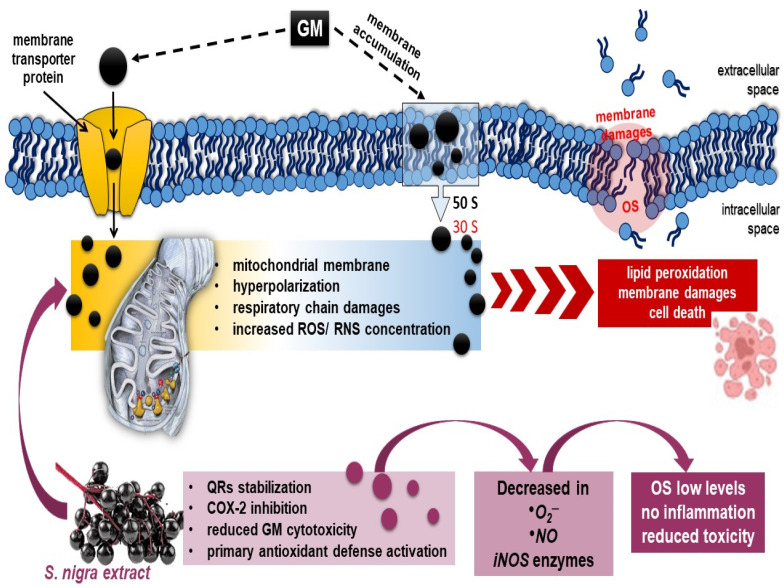
Possible mechanism of (1) GM-binding in 30S–50S ribosomal subunit and penetration by oxygen-dependent transport mechanism, hyperpolarization, and renal dysfunction [1,74]; (2) oral 7-day antioxidant compensatory response of *S. nigra*-lyophilized extract (containing rutin, epigallocatechin, and quercetin) in stimulation of renal mitochondrial dysfunction, improving renal tubular dynamics and halting the fibrotic process through complete ROS, NO•, and •O_2_^−^ generation neutralization and OS modulation [81,82,83]. As a sequence, *S. nigra*-lyophilized extract increases GM uptake and decreases GM nephrotoxicity. Abbreviations: GM, gentamycin; *S. nigra*; OS, oxidative stress; ROS, NO•, and •O_2_^−^, free radicals.

## 4. Materials and Methods

### 4.1. S. nigra Fruit Phytochemical Analysis and Ultrasound-Assisted Extraction

The *S. nigra* fruits were collected from Alino, Bulgaria, 2023 g. The juice was obtained by manual pressing and straining through sterile gauze and ultrasonically sonicated. For ultrasonic treatment (80 kHz), a bath with a piezoceramic emitter (Elmasonic P 30 H, Elma, Singen, Germany; volume 2.8 L; frequency 37–80 kHz; 22 °C) was used. The extract was filtered through cotton and a 0.8 μm nylon-membrane syringe filter (Acrodisc, Sigma-Aldrich, Sofia, Bulgaria). It has been established that with prolonged storage and an increase in pH and temperature, the anthocyanin content decreased. *S. nigra* fruits juice was subjected to lyophilization (300 rpm; 10 min; layer thickness 1 cm; −45 °C) and vacuum-sublimation drying in a TG 16.50 installation (Hochvacuum, Munich, Germany) for 24 ± 1 h [85,86].

### 4.2. High-Performance Liquid Chromatography (HPLC—DAD)

The high-performance liquid chromatography method (1100 HPLC, Agilent Technologies, Santa Clara, CA, USA) with a HPLC–DAD diode detector (G1315B, Agilent Technologies, Santa Clara, CA, USA) operated by an HP Chemstation was used to determine 9 components: (+)-epicatechin gallate; (-)-catechin; (-) epigallocatechin; rutin; quercetin; myricetin; kaempferol; and gallic and vanillic acids [87]. The optimized method was performed at 278 nm–368 nm wavelength, with a C18 reversed-phase column (Purospher star, Hiber RT 125-4; RP18, Purospher star, Merck, Sofia, Bulgaria). The new HPLC method is specific, sensitive, linear, and precise [88]. Separation was performed using a linear gradient elution program with 0.1% TCA (A) and 100% acetonitrile (B) for 40 min. The gradient elution program started with 5% B, 15% B at 16.5 min, 33% B at 22.5 min, 100% B at 30.5 min, and 5% B at 35 min until the 40th equilibration. The flow rate was 1.6 μL min^−1^, and the column temperature was −25 °C. The injection volume was set to 30 μL, and the DAD acquisition peaks were monitored simultaneously in the 200 ÷ 400 nm range. HPLC–DAD calibration curves were obtained by plotting the limits of the peak area against the standard concentrations (μg mL^−1^). All samples were filtered through a 0.46 µm acrodisc syringe filter before injection. Calibration curves consisted of five points, from 20 to 100 μg mL^−1^, for all analytics.

### 4.3. Antioxidant Ability In Vitro

The antioxidant ability of *S. nigra*-lyophilized extract was determined using a DPPH (2.2-diphenyl-1-picrylhydrazyl) assay, measured by the Electron Paramagnetic Resonance (EPR-X-Bruker-EMX^-micro^) spectroscopy [89]. Briefly, 80 mM DPPH ethanol solution was homogenized and incubated with *S. nigra*-lyophilized extract for 5 min (5 mM, 0.5 mg mL^−1^). The 0.20 µL was transferred in the Micro-221-EPR cavity at 23 °C. DPPH-H/R generation started immediately. The results were presented as % and recalculated as µg/DPPH radicals possible to be neutralized by 1 mL extract.

### 4.4. Lymphocyte Cultures and Chromosome Aberration Assay

Lymphocyte cultures were prepared from peripheral vein blood in heparin (30 IU mL^−1^) of healthy, nonsmoking, non-drinking (female), and drug-free donors aged 38 to 43 years.

Lymphocyte cultures (10 mL) were cultured in sterile flasks containing RPMI-1640 cell medium, 3 mL heat-inactivated normal calf serum, 0.2 mL re-substituted PHA (2%), and antibiotics (penicillin-gentamicin, 100 U mL^−1^: 50 μg mL^−1^) at 37 °C for 47 h. *S. nigra* extract (50 μL, 120 mg kg^−1^) was added to the lymphocytes at 37 °C for 47 h under aerobic conditions. After 47 h, the cells were incubated with colchicine solution (0.5 g mL^−1^) according to the classical culture protocol for 60 min. Finally, the culture medium was treated with 0.075 M KCl and 4 × methanol/acetic acid (3:1, *v*/*v*) at 23 °C, and the clastogenic effect was examined using the method of Evans, 1984 [90]. Microscopy of chromosomal metaphases stained with 10% Giemsa (Merck); chromosome aberration analysis was at ×1000 magnification (Olympus BX-41, Hamburg, Germany), with a minimum of 100 metaphases (Figure 2). Untreated lymphocyte samples were used as controls. All procedures followed the Declaration of Helsinki, approved by the Ethics and Academic Integrity Committee of Medical Faculty, Trakia University (No. 7.5.1 OD_4.1.5.9/protocol 18/15.04.2022). The lymphocyte cultures were stored in the Laboratory for Chromosomal Diagnostics and Genetic Monitoring and Screening in the Department of Molecular Biology, Immunology, and Medical Genetics, Medical Faculty, Trakia University, Stara Zagora, Bulgaria. All donors were informed about this experimental procedure and signed written informed consent forms.

### 4.5. Nephrotoxicity Induction and Therapeutic Protection

Thirty-six Balb/c mice (37–39.4 g; 9 weeks old, Neurobiology Institute, Slivnitsa, Bulgaria) used in this study were kept in the animal care facility by the Bioethics Committee, TrU, Stara Zagora, Bulgaria. Animals were maintained at a controlled temperature (21 °C), humidity (52%), and 12 h dark/light 10-day cycle of adaptive feeding and acclimatization, with a license (317/6000-0333/09.12.2021) following Directive 2010/63/EU on the animals’ protection used for experimental and other scientific work.

The GM-induced nephrotoxicity murine model was developed by a daily IP injection of 200 mg kg^−1^ day^−1^ for 10 consecutive days, according to previous models [7,15,24]. The GM inducement caused oxidative stress disturbances and generated a redox–homeostatic imbalance cascade, directly affecting the proximal renal tubules.

The *S. nigra* fruit extract has shown pharmacological properties at doses ranging from 15 to 600 mg kg^−1^ [91]. Its pharmacological effects may be different depending on the concentration of flavonoid and polyphenolic components. Based on our preliminary in vitro studies [92], the robust antioxidant activity and cytoprotective effect, 120 mg kg^−1^ *S. nigra* fruit extract was used in the present study.

The animals were divided into four groups (n = 6, Figure 2), according to the (1) controls; basal diet (19.6% protein, 4.03% fat, 6.89% fiber, 10.71% moisture; 8.97% ash) was injected IP with 1 mL ice-cold NaCl isotonic solution (0.9%); (2) *S. nigra* extract only administered per os (PO) (120 mg kg^−1^ day^−1^ bw); (3) GM only; GM was injected IP (200 mg kg^−1^ day^−1^ b) to induce acute nephrotoxicity; (4) GM + *S. nigra* therapy; GM was injected IP (200 mg kg^−1^ bw) and received PO *S. nigra*-lyophilized extract (120 mg kg^−1^ day^−1^ bw) after 2 h.

The *S. nigra* extract was mixed in cold NaCl isotonic solution (0.9%). The animals’ physiological state and behavior were monitored daily. Twenty-four hours after the last dose, the mice were weighed and anesthetized by IP injection (Nembutal, 50 mg kg^−1^). Blood samples collected by intra-cardiac technique in serum tubes were centrifuged (4000 rpm, 10 min, 4 °C) and analyzed immediately. The mice’s kidneys were weighed, and the left kidney was fixed in 10% formalin buffer for histological analysis. The right kidney was placed in ice-cold 0.05 M PBS (pH 7.5; 4 °C), homogenized individually, and examined.

### 4.6. Histopathological Analysis

The left kidney tissue was embedded in paraffin after being perfused, dehydrated by a graded series of ethanol, and fixed in 10% phosphate-buffered formalin for 24 h. The kidney tissues, cut into sections (5 µm), were mounted on gelatin-coated slides, xylene deparaffinized, and 0.1% hematoxylin–eosin (H&E) stain to distinguish significant kidney injuries such as tubular necrosis, fibrosis, inflammatory cells infiltration, tubular dilation, and cast formation.

### 4.7. Mast Cell (MCs) Number and Collagen Fiber Thickness (CFT)

Next, left kidney tissues were embedded in paraffin after being fixed in 10% formaldehyde/aqueous solution (7 days) and dehydrated. The kidney tissues were cut into slices (5 µm) and stained by toluidine blue (MCs differentiation) and Azan technique (collagen fibrils differentiation and indicating interstitial fibrosis). In short, paraffin sections were xylene deparaffinized, hydrated (ethanol 100%, ethanol 96%, ethanol, ethanol 80%, ethanol 70%), and washed in water. Toluidine blue (0.1%) in McLivane’s buffer (pH = 3) was used for metachromatic MCs [93], and the Azan stain was used for collagen fiber (CFT) visualization stained with blue, erythrocytes with orange, and muscle cells and nuclei with red [94].

### 4.8. Renal Hydroxyproline (HYP) u Protein Oxidation Analysis

Renal hydroxyproline analysis (at 110 °C for 24 h; hydrolysis with 6N HCl, incubation at 110 °C), used to quantify fibrotic changes, was determined at 550 nm absorption by the Woessner method [57] and expressed as μg/HYR per gram kidney tissue.

The protein oxidation analysis (albumin injuries) in kidneys was assessed by the EPR method in vivo, using spin-conjugation with spin-trap 3-maleimido proxyl (5-MSL). Right kidney tissue (10 mg) was mixed with 20 mM 5-MSL dissolved in 900 μL dimethyl sulfoxide (DMSO). The mixture was centrifuged (1000 rpm; 15 min) at 4 °C. The protein/albumin conformational (-SH) changes were recorded in triplicate, with the following parameters: 3505 G; 6.42 MW power; 5 G amplitude; 12 modulations, in random units, by the method described earlier [58].

### 4.9. Renal Functional Markers

To monitor renal functional damages, commercial kits to measure the kidney injury molecule-1 (KIM-1; No. MBS175125), cystatin C (CysC), glutathione-S-transferase (GST), and serum levels of gamma-glutamyl-transpeptidase (gamma-GT), creatinine (Cre), and urea (U) were used.

### 4.10. Oxidative Stress Analysis in Renal Tissue

To monitor renal antioxidant enzymes activities, catalase (CAT; No. ab83464), superoxide dismutase (SOD; No. ab65354), glutathione (GSH; No. ab142044), glutathione peroxidase 1 (GPx1; No. ab41464), as well as the pro-oxidants such as malondialdehyde (MDA; No. ab233471) were investigated by ELISA kits.

The ROS production: A total of 100 µL homogenized kidney tissue was mixed with 900 μL (50 mM) N-tert-butyl-alpha-phenylnitrone (PBN) dissolved in DMSO. The mixture was centrifuged at 4000× *g*, 10 min at 4 °C, by [95].

The nitric oxide (•NO) generation relative to the spin–adduct formed between the spintrap carboxy 2-(4-carboxyphenyl)-4,4,5,5-tetramethyl (CPTIO.K) and •NO in kidney tissue were based on established EPR methods [96,97]. Briefly, 50 μM CPTIO.K was dissolved in a mixture of 50 mM Tris (pH = 7.5), and DMSO (9:1) was centrifuged at 4000× *g* for 10 min at 4 °C. Then, 100 μL kidney samples were mixed in 100 μL CPTIO.K, and spin–adducts were recorded.

The superoxide (•O_2_^−^) concentration in kidney tissue was determined relative to the spin–adduct formed using the spin-trap CMH (1-hydroxy-3-methoxycarbonyl-2,2,5,5-tetramethylpyrrolidine), based on methods [98,99]. For this purpose, 30 µL kidney tissue was activated in 30 µL CMH (1:1) on an ice bath, and after 5 min incubation, it was prepared. All EPR analyses were performed with fivefold measurement in recorded EPR spectra, with the following characteristics: 3503–3515 G center field; 6.42–20.00 mW microwave power; 5–10 G modulation per sample, and the results are presented in arbitrary units (a.u.)

### 4.11. Pro-Collagen I Alpha 1, Heme Oxygenase-1, and Pro-Inflammatory Cytokines in Renal Tissue

The PGC-1, the pro-inflammatory IL-1, IL-10, IL-6, ITF-α, and TNF-γ cytokines were measured with ELISA kits.

### 4.12. Statistical Methods

The aberrance and aberrant cell degrees were processed by one-way ANOVA followed by χ2-Cramer’s V test (GraphPadPrism 6/Windows; GraphPad Software, Inc., USA). Data are presented as mean ± SD; *p* < 0.05 was considered statistically significant.

The mast cell (MCs) density (number/field of view × 400) was determined on a ×200 microscopic field with an area of 0.163 mm^2^ in kidney sections of each animal, using a light research microscope (LEIKA DM 1000, Leica Camera AG, Wetzlar, Germany) equipped with a digital camera (LEIKA DFC 290). The data were processed by one-way ANOVA followed by Tukey Kramer’s test (GraphPadPrism 6 for Windows; GraphPad Software, Inc., USA). Data are presented as mean ± SD; *p* < 0.05 was considered statistically significant. The remaining statistical analyses were performed using Excel version 10.0 software, StaSoft, Inc., San Diego, CA, USA, and presented as mean ± SD.

The EPR processing was performed using WIN-EPR *SimFonia* 1.2/6130860 software. Statistical analysis was performed using one-way ANOVA and Student’s *t*-test to determine differences; *p* < 0.05 was considered statistically significant.

### 4.13. Limits of This Study

The results related to our study clearly show that the lyophilized extract of *S. nigra* attenuated renal dysfunction caused by GM application. However, some limitations should be noted that could contribute to the overall significance of the extract on GM-induced nephrotoxicity. Due to ethical considerations, the number of animals studied per group was six, which was sufficient to reveal the most significant differences but might be insufficient to establish additional correlations between some parameters. The results indicate the involvement of some biomarkers, like KIM-1 and PGC-1α, in the process, but further investigation of markers through pathway-specific inhibitors or genetic approaches would better clarify their role. In the present study, in vivo GM activity was not investigated by the co-administration of *S. nigra* extract, which would also reveal additional information about the antimicrobial properties of the combination therapy. Further studies are needed to identify possible doses that have a beneficial effect on kidney intoxication with longer-term use in protecting ferroptosis processes and the molecular study of DNA damage.

## 5. Conclusions

The present study highlights the anti-inflammatory potential of *S. nigra*-lyophilized fruit extract (120 mg mL^−1^) by reducing fibrotic cell proliferation under GM-induced nephrotoxicity. The antioxidant protection is mediated by a reduction in cytokine expression and albumin modulation. The proposed mechanisms of action of *S. nigra* are directly related to the redox modulation of ROS, NO•, and •O_2_^−^, which facilitate mitochondrial antioxidant and anti-inflammatory properties, associated with the cytoprotection of GM-induced OS and reduced hyperpolarization. In conclusion, we encourage *S. nigra* use as a traditional supplement to restore antioxidant enzymes in patients with renal injury undergoing GM treatment without compromising the efficacy of bactericidal therapy.

## Data Availability

The original contributions presented in this study are included in the article. Further inquiries can be directed to the corresponding authors.

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
