# Peer review of "Sambucus nigra-Lyophilized Fruit Extract Attenuated Acute Redox–Homeostatic Imbalance via Mutagenic and Oxidative Stress Modulation in Mice Model on Gentamicin-Induced Nephrotoxicity"

_pharmaceuticals, 2025, doi:10.3390/ph18010085_

Round 1
Reviewer 1 Report
Comments and Suggestions for Authors
Dear Authors:
The study is well-conducted and explores an important topic. However, some grammatical and typographical issues need correction
-
Clarify and streamline repetitive sections, especially in the discussion (e.g., inflammatory marker modulation).
-
Provide additional justification for the chosen dose of S. nigra (120 mg/kg) and explain its relevance.
-
Simplify the abstract for better accessibility, emphasizing key results and their significance.
-
Add a brief mention of study limitations and future directions in the conclusion
-

Author Response
Reviewer 1
1.The study is well-conducted and explores an important topic. However, some grammatical and typographical issues need correction
The authors are grateful for the favorable evaluation of their work. We have tried to reduce all grammatical and typographical problems.
2. Clarify and streamline repetitive sections, especially in the discussion (e.g., inflammatory marker modulation).
We have streamlined repetitive sections, and some of the sub-paragraphs have been completely rewritten.
3. Provide additional justification for the chosen dose of S. nigra (120 mg/kg) and explain its relevance.
The S. nigra fruit extract has shown pharmacological properties at doses ranging from 15 - 600 mg kg-1 [94]. Its pharmacological effects may be different depending on the concentration of flavonoid and polyphenolic components. Based on our preliminary in vitro studies [Thesis, S. nigra extract] [95], the robust antioxidant activity and cytoprotective effect, 120 mg kg-1 S. nigra fruit extract was used in the present study.
Ref. 95. Thesis „Characterization and utilization of active antioxidants from black elder (S. Nigra extract)” Sofia, Bulgaria, 27.06.2022. Reference nomer: 0237
4. Simplify the abstract for better accessibility, emphasizing key results and their significance.
Done.
5. Add a brief mention of study limitations and future directions in the conclusion.
However, some limitations should be noted that could contribute to the overall significance of the extract on GM-induced nephrotoxicity. Due to ethical considerations, the number of animals studied per group was six, which is sufficient to reveal the most significant differences, but may be insufficient to establish additional correlations between some parameters. The results indicate the involvement of some biomarkers, such as KIM-1 and PGC-1α in the process, but further investigation of the markers by pathway-specific inhibitors or genetic approaches would better clarify their role. In the present study, in vivo GM activity was not investigated by co-administration of S. nigra extract, which would also reveal additional information about the antimicrobial properties of the combination therapy. Further studies are needed to identify possible doses that have a beneficial effect on renal intoxication with longer use; in the protection of ferroptosis processes and molecular study of DNA damage.

Reviewer 2 Report
Comments and Suggestions for Authors
The manuscript by Petkova-Parlapanska et al. investigates the protective effects of Sambucus nigra (S. nigra) lyophilized fruit extract against gentamicin (GM)-induced nephrotoxicity in mice. Gentamicin is a widely used antibiotic that can cause kidney damage as a side effect. The study explores whether S. nigra extract, known for its antioxidant and anti-inflammatory properties, can mitigate these toxic effect. The results are presented clearly and well-structured, using tables and figures to illustrate the data. However, greater clarity in the description of the graphs and tables could be helpful for the reader, especially for those not familiar with all the techniques used. For example, units of measurement should be indicated more clearly in all the figures. The introduction provides sufficient context for the study, presenting the use of gentamicin (GM) and its toxic side effects, including nephrotoxicity. The introduction describes the interest in exploring the effects of plants containing polyphenols and quercetin-like flavonoids in preventing inflammatory activation and reducing free radicals. However, a better transition between gentamicin toxicity and the introduction of S. nigra could be provided. It could be useful to explain more clearly how the antioxidant and anti-inflammatory action of S. nigra can counteract the toxicity mechanisms of gentamicin before introducing the specific study. The methods are described in detail, allowing for the reproducibility of the study. However, it would be useful to provide more details on the rationale for the doses used of S. nigra extract (120 mg/kg) and gentamicin (200 mg/kg). It would also be useful to better clarify the meaning of the unit "a.u." used for some measurements. Overall, the English language is understandable. However, there are some grammatical and stylistic inaccuracies. Some errors are noted in the use of verb tenses, subject-verb agreement, and the use of articles and prepositions. There are also some non-idiomatic expressions or unclear formulations. I suggest to add more detailed legends to the figures, with precise indications of the units of measurement and the treatments used. Strengthen the description of the numerical results in the text, directly linking them to the graphs and tables. Provide a smoother transition between the toxicity of gentamicin and the introduction of S. nigra, better explaining the rationale for using this plant to counteract the toxic effects of the drug. In addition it could be beneficial to deepen the discussion of the results, connecting them to previous studies and specifying the hypothesized mechanism of action of S. nigra in relation to the experimental observations. By following these suggestions, the manuscript could be significantly improved in terms of clarity, accuracy, and scientific impact.
Author Response
Reviewer 2
The manuscript by Petkova-Parlapanska et al. investigates the protective effects of Sambucus nigra (S. nigra) lyophilized fruit extract against gentamicin (GM)-induced nephrotoxicity in mice. Gentamicin is a widely used antibiotic that can cause kidney damage as a side effect. The study explores whether S. nigra extract, known for its antioxidant and anti-inflammatory properties, can mitigate these toxic effect.
The authors are grateful for the favorable evaluation of their work. We have tried to reduce all grammatical and typographical problems.
The results are presented clearly and well-structured, using tables and figures to illustrate the data. However, greater clarity in the description of the graphs and tables could be helpful for the reader, especially for those not familiar with all the techniques used. For example, units of measurement should be indicated more clearly in all the figures.
The authors are grateful for the comment. Corrections and a more in-depth description of the techniques, methods, and statistical analyses used have been made under each of the figures and tables.
The introduction provides sufficient context for the study, presenting the use of gentamicin (GM) and its toxic side effects, including nephrotoxicity. The introduction describes the interest in exploring the effects of plants containing polyphenols and quercetin-like flavonoids in preventing inflammatory activation and reducing free radicals. However, a better transition between gentamicin toxicity and the introduction of S. nigra could be provided. It could be useful to explain more clearly how the antioxidant and anti-inflammatory action of S. nigra can counteract the toxicity mechanisms of gentamicin before introducing the specific study.
Done.
The methods are described in detail, allowing for the reproducibility of the study. However, it would be useful to provide more details on the rationale for the doses used of S. nigra extract (120 mg/kg) and gentamicin (200 mg/kg). It would also be useful to better clarify the meaning of the unit "a.u." used for some measurements.
The S. nigra fruit extract has shown pharmacological properties at doses ranging from 15 - 600 mg kg-1 [94]. Its pharmacological effects may be different depending on the concentration of flavonoid and polyphenolic components. Based on our preliminary in vitro studies [Thesis, S. nigra extract] [95], the robust antioxidant activity and cytoprotective effect, 120 mg kg-1 S. nigra fruit extract was used in the present study.
Ref. 95. Thesis „Characterization and utilization of active antioxidants from black elder (S. Nigra extract)” Sofia, Bulgaria, 27.06.2022. Reference nomer: 0237
Corrections and additional clarification and description of the units of measurement used have been made in the text and under each of the figures and tables.
Overall, the English language is understandable. However, there are some grammatical and stylistic inaccuracies. Some errors are noted in the use of verb tenses, subject-verb agreement, and the use of articles and prepositions. There are also some non-idiomatic expressions or unclear formulations. I suggest to add more detailed legends to the figures, with precise indications of the units of measurement and the treatments used.
We have tried to reduce all grammatical and stylistic inaccuracies. Corrections and a more in-depth description of the techniques, methods, and statistical analyses used have been made under each of the figures and tables.
Strengthen the description of the numerical results in the text, directly linking them to the graphs and tables. Provide a smoother transition between the toxicity of gentamicin and the introduction of S. nigra, better explaining the rationale for using this plant to counteract the toxic effects of the drug.
Done.
In addition it could be beneficial to deepen the discussion of the results, connecting them to previous studies and specifying the hypothesized mechanism of action of S. nigra in relation to the experimental observations. By following these suggestions, the manuscript could be significantly improved in terms of clarity, accuracy, and scientific impact.
Following the above suggestions, our proposed manuscript was significantly improved in terms of clarity, accuracy, and scientific impact.

Reviewer 3 Report
Comments and Suggestions for Authors
The manuscript investigates the therapeutic potential of Sambucus nigra lyophilized fruit extract (S. nigra extract) in relieving gentamicin (GM)-induced nephrotoxicity in a mouse model. The study provides significant insights into the antioxidant, anti-inflammatory, and anti-fibrotic effects of S. nigra extract, elucidating its underlying mechanisms of action. The methodology, results, and discussion are well-organized, but some areas require clarification and refinement.
· While the lyophilization parameters are mentioned, the justification for selecting the dose (120 mg/kg/day) is missing.
· The role of specific bioactive compounds in S. nigra extract (e.g., rutin, epigallocatechin) should be better linked to observed therapeutic effects. Are there any additive or synergistic effects between these compounds?
· The manuscript hypothesizes about the involvement of KIM-1 and PGC-1α pathways but lacks experimental data to confirm these mechanistic claims. Inclusion of pathway-specific inhibitors or genetic approaches would strengthen the conclusions.
· While multiple comparisons are presented, the specific statistical tests used for each dataset are not always clear. A summary table of statistical methods should be included
· Figure 1 HPLC-DAD chromatograms and Figure 5 histopathological images lack sufficient resolution and annotations for clear interpretation. Adding arrows or labels to highlight key findings would improve readability.
· Certain sentences are verbose and could be streamlined for clarity. For example, "The findings suggest that S. nigra attenuates renal dysfunction and structural damages by modulating oxidative stress and acute inflammation" could be simplified.
· Some technical terms (e.g., "anticlastogenic") may require brief definitions for a broader audience.
Overall, the manuscript is scientifically valuable and has the potential to contribute significantly to the field. Addressing the above concerns will enhance the robustness, clarity, and impact of the study. Specific attention should be given to providing mechanistic insights, refining statistical analyses, and improving the presentation of data.
Author Response
Reviewer 3
The manuscript investigates the therapeutic potential of Sambucus nigra lyophilized fruit extract (S. nigra extract) in relieving gentamicin (GM)-induced nephrotoxicity in a mouse model. The study provides significant insights into the antioxidant, anti-inflammatory, and anti-fibrotic effects of S. nigra extract, elucidating its underlying mechanisms of action. The methodology, results, and discussion are well-organized, but some areas require clarification and refinement.
The authors are grateful for the favorable evaluation of their work.
- While the lyophilization parameters are mentioned, the justification for selecting the dose (120 mg/kg/day) is missing.
The S. nigra fruit extract has shown pharmacological properties at doses ranging from 15 - 600 mg kg-1 [94]. Its pharmacological effects may be different depending on the concentration of flavonoid and polyphenolic components. Based on our preliminary in vitro studies [Thesis, S. nigra extract] [95], the robust antioxidant activity and cytoprotective effect, 120 mg kg-1 S. nigra fruit extract was used in the present study.
Ref. 95. Thesis „Characterization and utilization of active antioxidants from black elder (S. Nigra extract)” Sofia, Bulgaria, 27.06.2022. Reference nomer: 0237
- The role of specific bioactive compounds in S. nigra extract (e.g., rutin, epigallocatechin) should be better linked to observed therapeutic effects. Are there any additive or synergistic effects between these compounds?
Presumably, the 7-day-lasting antioxidant compensatory response of S. nigra, containing rutin, epigallocatechin, quercetin, stimulated Th1/Th2 response and protects renal mito-chondrial dysfunction by improving renal tubular dynamics and arresting the fibrosis process [86]. Moreover, the potential of the synergistic phytochemicals rutin, epigallocat-echin, quercetin and myricetin in S. nigra extract is expressed in the complete antagonism/ HO•, •O2−, NO• scavenging, inhibition of lipid peroxidation and complete oxidative stress modulation. It has been shown that the synergistic phytochemicals functionally have a ROS/ PNH scavenging capacity 100–300 times higher than the mannitol [10, 87]. S. nigra extract disrupted of extracellular matrix components that allow full penetration of antibiotic, enhance their effect, and simultaneously prevents the accumulation of acute renal dysfunction.
- The manuscript hypothesizes about the involvement of KIM-1 and PGC-1α pathways but lacks experimental data to confirm these mechanistic claims. Inclusion of pathway-specific inhibitors or genetic approaches would strengthen the conclusions.
The proposed hypothesizes has been reorganized. The results indicate involvement of some biomarkers, like KIM-1 and PGC-1α in the pro-cess, but further investigation of markers through pathway-specific inhibitors or genetic approaches would better clarify their role. In the present study, in vivo GM activity was not investigated by the co-administration of S. nigra extract, which would also reveal additional information about the antimicrobial properties of the combination therapy. Further studies are needed to identify possible doses that have a beneficial effect on kidney intoxication with longer-term use; in protecting ferroptosis processes and the molecular study of DNA damage.
- While multiple comparisons are presented, the specific statistical tests used for each dataset are not always clear. A summary table of statistical methods should be included.
Done.
- Figure 1 HPLC-DAD chromatograms and Figure 5 histopathological images lack sufficient resolution and annotations for clear interpretation. Adding arrows or labels to highlight key findings would improve readability.
Done.
- Certain sentences are verbose and could be streamlined for clarity. For example, "The findings suggest that S. nigra attenuates renal dysfunction and structural damages by modulating oxidative stress and acute inflammation" could be simplified.
Done.
- Some technical terms (e.g., "anticlastogenic") may require brief definitions for a broader audience.
Done.
- Overall, the manuscript is scientifically valuable and has the potential to contribute significantly to the field. Addressing the above concerns will enhance the robustness, clarity, and impact of the study. Specific attention should be given to providing mechanistic insights, refining statistical analyses, and improving the presentation of data.
The authors are grateful for the comments made, which enrich, systematize, and raise the level of the proposed manuscript.
